# Morphological, Pathogenic and Molecular Characterization of *Sclerotinia sclerotiorum*, the Causal Agent of White Rot of Cabbage (*Brassica oleracea* var. *capitata*), in Serbia

**DOI:** 10.3390/plants14162478

**Published:** 2025-08-10

**Authors:** Brankica Pešić, Petar Mitrović, Ana Marjanović Jeromela, Federica Zanetti, Milica Mihajlović, Jovana Hrustić, Mira Vojvodić, Miljan Grkinić, Aleksandra Bulajić

**Affiliations:** 1Institute of Pesticides and Environmental Protection, Banatska 31b, 11080 Belgrade, Serbia; brankica.pesic@pesting.org.rs (B.P.); milica.mihajlovic@pesting.org.rs (M.M.); jovana.hrustic@pesting.org.rs (J.H.); 2National Institute of Field and Vegetable Crops, Maksima Gorkog 30, 21000 Novi Sad, Serbia; petar.mitrovic@ifvcns.ns.ac.rs (P.M.); ana.jeromela@ifvcns.ns.ac.rs (A.M.J.); 3Department of Agricultural and Food Sciences (DISTAL), Alma Mater Studiorum—University of Bologna, Piazza Goidanich 60, 47521 Cesena, Italy; federica.zanetti5@unibo.it; 4Faculty of Agriculture, University of Belgrade, Nemanjina 6, 11080 Belgrade, Serbia; miravojvodic2510@gmail.com (M.V.); beomiljan@gmail.com (M.G.)

**Keywords:** morphological characterization, phylogeny, haplotype diversity, aggressiveness, Brassicaceae susceptibility

## Abstract

Sclerotinia sclerotiorum is a globally distributed necrotrophic pathogen with a broad host range, including many Brassicaceae crops. In 2021, white rot symptoms on cabbage were observed in 12 commercial fields in the northern part of Serbia. Twelve representative isolates of *S. sclerotiorum*, forming white colonies and black sclerotia, were selected for characterization and comparison with an isolate from sunflower, as the most important host plant in Serbia. The isolates showed significant variation in growth rate and sclerotia production, while ITS sequence analysis revealed the complete nucleotide identity and all isolates grouped within the major phylogenetic clade of *S. sclerotiorum*. Despite the low diversity of the global population of *S. sclerotiorum*, forty-four haplotypes were identified, with one predominant haplotype encompassing all Serbian isolates. When six Brassicaceae species (cabbage, cauliflower, broccoli, kale, mustard, and oilseed rape) and sunflower were inoculated, sunflower was found to be the most and cauliflower the least susceptible, while isolates from cabbage were generally more aggressive than those derived from sunflower. This work represents the first detailed characterization of *S. sclerotiorum* infecting cabbage in Serbia and highlights its genetic uniformity and differential pathogenic potential, which are critical factors for integrated disease management and crop rotation planning in Brassicaceae agroecosystems.

## 1. Introduction

Cruciferous plants (family Brassicaceae) are an important group of oilseeds and vegetable crops that make an important contribution to the economy in many countries around the world. They represent a large share of oil and vegetable production, as well as participating in a growing biofuel industry. Cabbage (*Brassica oleracea* var. *capitata*) is the most important cruciferous vegetable. Cabbage production varies worldwide, with China being the leading producer (34,986,293 tons), followed by India (9,825,000 tons), the Republic of Korea (2,428,893 tons) and Russia (2,298,209 tons). In Serbia, it is grown on 7111 ha (FAO, https://www.fao.org/faostat/en/#data/QCL, accessed on 24 May 2025), mainly in the central and western parts of Serbia (42%), followed by the southern and eastern regions (25%) [1]. Several diseases threaten cabbage production, especially diseases caused by fungi such as leaf spot, downy mildew, damping-off, *Sclerotinia* rot/white rot, yellow rot or *Fusarium* wilt, blackleg, wire stem and clubroot. Two related *Sclerotinia* species, *Sclerotinia sclerotiorum* [2] and *S. minor* [3], have been described as causal agents of *Sclerotinia* rot in cabbage and other Brassicaceae plants. Depending on weather conditions, the yield losses caused by both species vary between 5 and 100% in many crops [4]. The average annual incidence of *Sclerotina* rot in oilseed rape in China is about 10–20% and can be as high as 80% in severe outbreaks [5]. Significant losses due to *S. sclerotiorum* in oilseed rape (*Brassica napus*) and mustard (*Brassica juncea*) have been recorded in Australia, North America, China and Europe [6]. *Sclerotinia* species attack all growth stages of the crop and cause commercial yield losses in the millions under field, storage and marketing conditions. The reported rot incidence caused by *S. sclerotiorum* in cabbage varies between 1 and 30% [7,8,9,10,11].

*S. sclerotiorum* is one of the most devastating soilborne pathogens affecting many agricultural crops [12,13] and is distributed in more than 92 countries all over the world (CABI distribution map, UK, https://plantwiseplusknowledgebank.org, accessed on 29 May 2025). The pathogen has an extraordinarily broad host range within more than 98 families and is associated with 2048 host species and varieties (USDA database, https://biocollections.ars.usda.gov, accessed on 29 May 2025). Most of the hosts are dicotyledonous herbaceous species, but some also belong to monocotyledonous plants. In Serbia, *S. sclerotiorum* is still rarely studied. Significant losses have been recorded in sunflower production [14], and the occurrence in fields with green beans (*Phaseolus vulgaris*) [15], faba beans (*Vicia faba*) [16] and cabbage [17] has been observed. However, in these studies, the pathogen was not isolated and subsequently characterized, so data on pathogenic, morphological and genetic characteristics of isolates from Serbia are missing. Despite the fact that *S. sclerotiorum* has been described in cabbage and other Brassicaceae vegetables worldwide [2,3,18], its distribution and importance in cabbage production in Serbia had not been investigated prior to this study.

In 2021, typical *Sclerotinia* rot-like symptoms were observed on cabbage plants in several commercial fields in the northern part of Serbia, including spreading brown lesions and intense rotting of cabbage heads, often accompanied by abundant snow-white mycelium and black sclerotia (Mitrović, personal communication). The aim of the present work was to (a) identify and characterize the causal agent of the observed symptoms on cabbage based on pathogenic and morphological features; (b) determine phylogenetic relationships among isolates; (c) investigate intraspecific diversity by comparing isolates from cabbage and sunflower; (d) determine the population diversity of *S. sclerotiorum* worldwide and in Serbia based on haplotype analyses; (e) test the aggressiveness of derived isolates against cabbage and five other Brassicaceae species and compare them with sunflower as the main *Sclerotinia* host plant in Serbia; and (f) determine the relative differences in susceptibility of the main Brassicaceae crops.

## 2. Results

### 2.1. Disease Symptoms and Morphological Characterization

Typical symptoms of the *Sclerotinia* rot outbreak were observed in 12 production fields at two locations, Begeč and Futog, mostly distributed in patches. The first symptoms of the disease were water-soaked, spreading areas on the lower leaves, which became necrotic and were covered with a fluffy white mycelium (Figure 1A,B). In the late stages of disease progression, black sclerotia 2 to 7 mm in size were detected, either embedded in the white mycelium (Figure 1D) or located in the decaying outer leaves of the cabbage heads (Figure 1C). As the infection progressed, head rot and complete collapse of the plants were observed. The incidence of the disease varied slightly among fields. The highest incidence was observed in the field from which isolate M originated and was estimated at 10%, followed by the field from which isolate M3 originated (3%). In the remaining fields, the disease incidence ranged from 1 to 2%.

From the collected symptomatic cabbage samples, 12 *Sclerotinia*-like isolates were obtained, 4 isolates from Begeč (B, B1, B2, and B3) and 8 isolates from Futog (M, M1, M2, M3, M4, M5, M6, and M7), with 1 representative per cabbage production field, all of which showed a similar colony appearance, typical for *Sclerotinia* spp. Initially, the isolates formed white, fast-growing colonies with uniform margins consisting of sparse, homogeneous mycelium adhering to the medium (Figure 1E,H). After filling the plates, the isolates developed sparse to very dense, floccose, wooly–floccose or wooly aerial mycelium containing tufts and strands, usually arranged in a circular pattern (Figure 1I–L). The growth rate of the isolates varied between 25.3 mm/day (isolate M4) and 35.4 mm/day (isolate B), with an average of 30.1 mm/day. The growth rate of isolate SC from sunflower was 25.6 mm/day. The differences among the isolates in terms of average growth rate were statistically significant (F = 31.1; *p* < 0.0001) (Figure 2). On the surface of the four-day old colony, all cabbage-derived isolates formed white sclerotial initials that developed to grayish-black, mostly solitary sclerotia after an incubation period of 6–9 days (Figure 1F,D). The majority of isolates from cabbage formed sclerotia after 8 days (Figure 1G). In the culture of the sunflower-derived SC isolate, sclerotia were visible much later, 13 days after inoculation (Table 1). All sclerotia were round or irregularly shaped, had a textured surface and were easily detached from the mycelium. Based on the arrangement of sclerotia in the cultures, the isolates formed two distinct groups—group I, which formed a ring of individual sclerotia (5 isolates) mostly located at the edge of the colony, and group II, which formed two rings of sclerotia (edge and middle ring) (8 isolates). The isolate from the sunflower belonged to group II. The number of sclerotia formed differed significantly among the isolates from cabbage (*p* < 0.01). After 30 days of incubation, the average number of sclerotia varied between 9.7 in isolate M2 and 33.7 sclerotia/plate in isolate M7. During the period studied, isolate SC from sunflower formed a significantly lower number of sclerotia (2.7 sclerotia/plate) (Table 1).

### 2.2. Pathogenicity

The first symptoms of necrosis and decay were visible on the inoculated cabbage and sunflower plants 3 days post inoculation, and none of the plants inoculated with sterile agar plugs developed symptoms. All isolates were successfully recovered from infected tissue and thus fulfilled Koch’s postulates.

### 2.3. Sequence Analysis and Phylogeny

All 13 Serbian *S. sclerotiorum* isolates (12 from cabbage and 1 from sunflower) had an identical ITS nucleotide sequence (identity of 100% and 0 bp differences). BLAST (BLAST+ 2.17.0) analyses showed that they had 100% nucleotide identity with over 50 sequences of *S. sclerotiorum* in the GenBank database, and the most similar was isolate *S. sclerotiorum* from South Korea from Chinese chives (KJ614564) [19]. Phylogenetic analyses of the ITS sequences of 13 Serbian isolates, together with the previously listed sequences of *S. sclerotiorum, S. trifloriorum* and *S. minor,* resulted in a stable phylogenetic three with clear separation of all three *Sclerotinia* spp. (Figure 3). All 13 isolates from Serbia clustered in a well-supported branch comprising all *S. sclerotiorum* isolates with low diversity and were clearly separated from *S. minor* or *S. trifoliorum*, confirming the conventional identification.

Therefore, based on the morphology and growth characteristics of the recovered isolates, reproduced symptoms on inoculated cabbage and sunflower plants, sequence and phylogenetic analysis, the causal agent of the investigated cabbage disease was identified as *S*. *sclerotiorum*.

### 2.4. Haplotype Structure and Genetic Diversity of Sclerotinia sclerotiorum Sequences

The dataset of 1052 *S. sclerotiorum* sequences, including Serbian ones, was grouped into 44 haplogroups and characterized by 94 variable positions. The haplotype diversity of 0.102, as well as the nucleotide diversity of 0.00520, indicates a low level of genetic variation among the analyzed sequences (0 to 40 nt difference). The most numerous and dominant haplotype (Hap1) contained 997 sequences from different hosts and countries. The remaining 43 haplogroups (Hap2–44) comprised 55 sequences. The Serbian population of 13 *S. sclerotiorum* isolates was grouped into Hap1, with 0 nt difference between them. The haplotype network reconstructed using the median joining network algorithm implied star-shaped genealogical relationships with Hap1 in the central position (Figure 4) and numerous single-step mutations leading to peripheral haplotypes in the world population of *S*. *sclerotiorum*. The network also exhibited some genetic diversity, as evidenced by multiple mutational steps between specific haplotypes.

### 2.5. Aggressiveness Towards Different Plant Species

After a three-day incubation period, all inoculated plants of seven tested plant species showed typical symptoms of *Sclerotinia* rot (Figure 1M,N), while none of the control plants inoculated with sterile agar plugs developed symptoms. Artificial inoculation revealed significant differences between the *S. sclerotiorum* isolates in terms of their aggressiveness towards a particular plant species, as well as differences in the aggressiveness of a particular isolate towards different plant species (*p* < 0.05) (Figure 1O, Figure 5, Figure 6, Figure 7 and Figure 8). On average, isolate M3 was the most aggressive for all hosts tested, causing a cumulative disease severity index value of 19.4 (Figure 8). It was the most aggressive to sunflower and least aggressive to kale. The least aggressive isolate overall was isolate SC from sunflower, with a cumulative disease severity index value of 7.3 (individual values ranged from 0.4 to 2.2, depending on the plant species), followed by isolate M4, which was the least aggressive isolate from cabbage (cumulative disease severity index value 11.5, individual values 0.6 to 3.7). A susceptibility profile of the hosts to all isolates tested showed that sunflower was the most susceptible species (disease severity index values 2.2 to 3.9, cumulative 43.5), followed by cabbage (disease severity index between 1.4 and 3.6, cumulative 29.0), while cauliflower had the lowest susceptibility, with disease severity index median values between 0.1 and 2.4 (Figure 5, Figure 6 and Figure 7), depending on the isolate. The cumulative disease severity index for cauliflower was 15.6, confirming its low susceptibility to the isolates tested (Figure 7).

## 3. Discussion

As far as the authors know, this is the first study reporting a comprehensive characterization of *S. sclerotiorum* as the causal agent of cabbage rot in Serbia. *S. sclerotiorum* is a globally recognized causal agent of cabbage white rot, but in Serbia the symptoms were detected for the first time in 1997, published in 2016 [17], and no further research has been conducted to investigate its population. The occurrence of *S. sclerotiorum* on other host plants in Serbia, especially in the province of Vojvodina, has been documented previously [14]. In this study, we characterized a population of *S. sclerotiorum* responsible for substantial yield losses in cabbage and evaluated the relative susceptibility of six other crucifers to contribute to the development of a knowledge-based crop rotation strategy with crucifers. Our study is the first comprehensive characterization of the *S. sclerotiorum* population affecting cabbage in Serbia that offers valuable guidance for cabbage growers, particularly in selecting appropriate Brassicaceae crops for crop rotation.

Following an outbreak of cabbage white rot at 12 locations, the pathogen was identified as *S. sclerotiorum* using morphological and molecular methods. All isolates formed white colonies accompanied by large black sclerotia typical of *S. sclerotiorum*, as previously described [20]. The isolates in this study showed a substantial diversity in the appearance of the aerial mycelium, which ranged from sparse to very dense and from wooly to woolly–floccose or floccose in terms of its pattern on the colony surface. The morphological diversity of *S. sclerotiorum* isolates has been recorded in different crops [21]. Although the variability of colony color, ranging from white to grayish white, has been noted in many studies [21,22,23,24,25,26], our isolates shared the predominant appearance of *S. sclerotiorum* and formed exclusively white colonies [20,27,28,29,30]. The growth rate of the isolates examined was also within the range of previously reported values [22,23,26,31], although statistically significant differences were observed among the isolates.

All isolates from our study originating from cabbage formed roundish or irregular sclerotia after an incubation period of 6 to 9 days, similar to the isolates originating from *B. juncea* in India [32]. The previously reported time required for the formation of sclerotia in isolates from soybean ranged from 10.7 to 18 days [25], and 4 to 7 days in isolates from beans [29]. In the present study, the isolate from sunflower formed sclerotia much later (after 13 days). The number of sclerotia formed by isolate SC was also much lower (2.7 sclerotia/plate) compared to isolates from cabbage, which formed 9.7–33.7 sclerotia/plate. No clear correlation was found between the number of sclerotia and the time required for their formation.

The phylogenetic studies based on the ITS sequence allowed for the reliable identification of all isolates from Serbia, which clustered in a well-defined and supported branch that included all *S. sclerotiorum* isolates and was clearly separated from *S. minor* or *S. trifoliorum*. Our data are consistent with the previously published results [33]. A similar study based on phylogenetic analyses of ITS sequences of *S*. *sclerotiorum* reported the genetic diversity of 65 *S. sclerotiorum* isolates derived from *B. juncea* in India and described the presence of 11 evolutionary lineages [32]. We compared the Serbian isolates and the isolates from that study, and all isolates shared 100% nucleotide similarity (13 Serbian and 64 Indian), with only one isolate (Acc. No. MF408249) showing a 3 nt difference. Thus, we were not able to confirm the presence of diversity and correlate it with the different aggressiveness of Serbian or Indian isolates derived from cabbage or Indian mustard as two Brassicaceae crops. In our study, the susceptibility of mustard as a host plant was slightly lower than that of cabbage, which is the second most susceptible host plant. The Serbian isolates showed differences in aggressiveness, similar to the population of mustard-originating *S. sclerotiorum* from India [32], but a correlation with molecular diversity could not be established.

The molecular ITS marker is known as a useful barcoding region for the genus identification of fungi, as well as for the species delimitation of several robust genera such as *Sclerotinia* [33,34] and *Monilina* [35,36]. Our analyses of the haplotype structure and diversity of *S. sclerotiorum* revealed considerable genetic uniformity in the population of all sequences available to date. The analyzed set consists of 44 haplotypes, one of which is a major haplotype that accounts for almost 95% of all analyzed sequences and is distributed throughout the world. The central position of the major haplotype in a star-shaped genetic structure indicates its possible role as an ancestor. On the other hand, several haplotypes (e.g., Hap14 and Hap38) were found to be more distantly connected, suggesting possible isolated evolutionary lineages. After analyzing a smaller set of IGS sequences of *S*. *sclerotiorum,* a similarly low level of genetic structuring in the population was established [37]. These findings support the hypothesis that *S. sclerotiorum* isolates represent a genetically uniform population in different regions, with limited intra-population variability. Further studies, using diverse and additional molecular markers, are essential to clarify the extent and biological relevance of this genetic homogeneity.

To assess the potential impact of cross hosts within the crop rotation and the associated agronomic risks, we investigated the variability in aggressiveness of isolates against six Brassicaceae hosts and sunflower, an economically important crop in Serbia and a major host of S. *sclerotiorum*. To ensure methodological consistency and comparability, lesion length was selected as the primary indicator of susceptibility to *S. sclerotiorum,* a parameter well-established in the literature, especially in controlled inoculation experiments [5,6,14,21,22], and applicable across plant species with distinct anatomical and morphological features. Artificial inoculation revealed significant differences in the aggressiveness of *S. sclerotiorum* isolates. All isolates elicited symptoms similar to those observed in the field and described by many authors. In general, they were much more aggressive towards sunflower than all the other crucifers species tested. Among the latter, cauliflower was the least susceptible and cabbage the most susceptible species. A few studies have been published describing a great difference in the susceptibility of different crops to *Sclerotinia* species, from a study detecting no significant differences between cruciferous crops [38] to study showing some differences between Cucurbitaceae vegetables crops [39]. Based on differences in aggressiveness towards oilseed rape, *S. sclerotiorum* isolates were divided into three groups: highly aggressive, aggressive and slightly aggressive [40]. A similar categorization was reported by some other studies [21,25,32,41]. In the present study, all isolates were highly aggressive towards sunflower, while the statistically significant differences in aggressiveness towards Brassica species were biologically rather moderate. When analyzing the diversity of *S. sclerotiorum* isolates derived from *B. juncea* in India, no correlation between genetic variability and aggressiveness or geographical distribution was observed [32]. In the present study, the aggressiveness of each *S. sclerotiorum* isolate was largely determined by the specific Brassica host plant. Although isolate M3 was not the most aggressive for all plant species tested, it exhibited the highest cumulative aggressiveness across all plant species, warranting further investigation of the factors involved in *S. sclerotiorum* pathogenicity. Conversely, the isolate from sunflower showed significantly lower aggressiveness towards all tested host plants compared to the isolates from cabbage. As only one isolate from sunflower was included in this study, further studies with more isolates from sunflower are required to draw a comprehensive conclusion. Nevertheless, our results indicate a remarkable aggressiveness of isolates from cabbage towards sunflower, highlighting potential problems in crop rotation strategies. Pronounced phenotypic variability in *S. sclerotiorum* isolates, particularly regarding aggressiveness and host specificity, suggests that molecular markers alone may not fully capture the pathogen’s adaptive potential. To unravel the mechanisms underlying phenotypic variability, future research should focus on multi-locus genotyping or whole-genome sequencing, coupled with long-term field surveillance, to detect the emergence of highly aggressive or host-adapted strains. In Serbia, crop rotation is strictly adopted and in all the production fields considered in the present study, cabbage was grown either for the first time or as part of a 3–5 year rotation with soybean, potato and wheat, suggesting possible problems with other included crops. The combined airborne and soilborne nature of *S. sclerotiorum* and its global economic importance in many crops require detailed characterization of the pathogen in order to develop effective control measures. To the best of our knowledge, this study provided the first detailed description of the morphological, pathogenic and phylogenetic characteristics of *S. sclerotiorum* isolates from cruciferous plants in Serbia. Variations were observed in the growth rate, relative density and appearance of areal mycelium, as well as in aggressiveness. On average, cauliflower and kale were the least and cabbage the most susceptible Brassica species to the isolates tested. Compared to sunflower, all Brassica species were significantly less susceptible. Although the results of this study provide important insights into the aggressiveness of *S. sclerotiorum* isolates from cabbage in Serbia, these tests were carried out under controlled conditions which do not fully reflect the complex interactions between the pathogen, host plant, and environmental factors that influence disease development in agroecosystems. Nevertheless, this study provides a valuable foundation for understanding the presence and biological behavior of *S. sclerotiorum* in cabbage production, as well as guidance for future research aimed at improving sustainable disease management strategies.

The present study clearly revealed that even genetically homogeneous populations of *S. sclerotiorum* can display considerable phenotypic plasticity, likely driven by environmental conditions and host-specific selective pressures. The elucidation of pathogenic variability within this clonal population highlights the need for incorporating both molecular and phenotypic characterization in the development of integrated disease management strategies in Brassicaceae cropping systems.

## 4. Materials and Methods

### 4.1. Sampling and Pathogen Isolation

In September 2021, cabbage plants with head necrosis were observed in 8 fields near the locality of Futog (seven fields with cv. Futoški and one with cv. Bravo F1 hybrid) and 4 fields near Begeč (two fields with cv. Bravo F1 hybrid, one with cv. Bucharest and one with cv. Futoški), all in the province of Vojvodina, Serbia. The observed disease outbreak did not appear to be directly influenced by the weather conditions, since they were typical for that region and the time of year. At each location, disease incidence was estimated by walking the field in a zigzag course and randomly assessing 100 plants in three replicates. A total of 10 symptomatic samples were collected from each location and transported to the laboratory for isolation and identification. For each sample, small fragments of tissue were taken from the border between necrotic and healthy cabbage tissue, surface-sterilized with 2% sodium hypochlorite for 1 min, rinsed with sterile distilled H_2_O, plated on potato dextrose agar (PDA; 200 g potato, 20 g dextrose, 17 g agar, and 1 L distilled H_2_O) [42], and incubated at 24 °C for 5 days. Uniform *Sclerotinia*-like colonies grew from diseased cabbage tissue, and representative isolates from each field were purified by transferring the hyphal tips to PDA and selected for morphological identification and characterization. Isolates were stored on sealed PDA slants at 4 °C in the fungal collection of the Department of Phytopathology, Faculty of Agriculture, University of Belgrade. In addition, a reference isolate SC from sunflower from Rimski Šančevi, Province of Vojvodina, Serbia, was included in the study (fungal collection of the Institute of Vegetable and Field Crops, National Institute of the Republic of Serbia).

### 4.2. Morphological Identification

Morphological identification of 12 selected *Sclerotinia*-like isolates from cabbage and a reference isolate from sunflower was based on the colony appearance assessed 15 days after inoculation on PDA at 24 °C in darkness. The growth rate was determined by measuring two perpendicular colony diameters in five replicates per isolate and calculating an average value for each isolate. The presence of sclerotia and the time of first formation were assessed in the first week after subculturing, the distribution of sclerotia was evaluated after 15 days, and the number of sclerotia per colony and the size of sclerotia were determined 30 days after subculturing [43]. Three replicates were used for each isolate and the entire experiment was performed twice.

### 4.3. Amplification and Sequencing of Isolates’ DNA

Total genomic DNA was extracted from 100 mg dry mycelium of 7-day-old cultures of *Sclerotinia* isolates grown on potato dextrose broth (PDB; 200 g potato, 20 g dextrose and 1 L distilled H_2_O) using the DNeasy Plant Mini Kit (Qiagen, Hilden, Germany), according to the manufacturer’s instructions. PCR amplification of ITS (ITS1, 5.8S rDNA and ITS2) with the primers ITS1f/ITS4 [44,45] was performed in a total reaction volume of 25 μL, consisting of 12.5 μL 2 X PCR Master mix (Fermentas, Lithuania), 6.5 μL RNase-free water, 2.5 μL forward and reverse primers (working solution with a final concentration of 100 pmol/μL, Metabion International, Gräfelfing, Germany) and 1 μL template DNA. The amplification conditions were as follows: initial denaturation at 95 °C for 3 min, followed by 35 cycles of denaturation at 95 °C for 30 s, annealing at 52 °C for 1 min, elongation at 72 °C for 1 min and final elongation for 10 min at 72 °C. The amplicons obtained were stained with ethidium bromide, analyzed by 1% agarose gel electrophoresis and visualized using a UV transilluminator. The PCR products were sequenced directly in both directions using an automatic sequencer (Automatic Sequencer Macrogen Inc., Amsterdam, The Netherlands) using the same primers as for amplification. The consensus sequences were calculated with ClustalW 2.1 [46], integrated into the software MEGA 7.0 [47] and deposited in GenBank (http://www.ncbi.nlm.nih.gov, accessed on 7 August 2025). All generated sequences were compared with each other by calculating nucleotide (nt) similarities and with previously deposited isolates of *Sclerotinia* spp. in GenBank using the BLAST similarity search tool.

### 4.4. Phylogenetic Analyses

Newly generated Serbian ITS sequences of *Sclerotinia* isolates from cabbage were analyzed with 24 previously listed type species of the genus *Sclerotinia* [33,48,49] and an outgroup taxa *Trichoderma lixii* [50]. Of the 25 sequences retrieved in GenBank, 15 were *S. sclerotiorum* isolated from different host plants and geographical distributions. The analyses included five representative sequences of *S. trifloriorum* and four of *S. minor* (Table 2). A phylogenetic tree was constructed using the Maximum Likelihood Method implemented in MEGA 7.0 software [47]. The Gamma-distributed Kimura’s two-parameter model [51] obtained using the model test implemented in MEGA 7.0 was used as the best fitting model for nucleotide substitution, and all sites with gaps were omitted. The reliability of the obtained trees was evaluated with 1000 bootstrap replicates.

### 4.5. Haplotype Analysis of Sclerotinia sclerotiourum Sequences

All 1100 available sequences of the ITS region of *S. sclerotiorum* (accessed 25 May 2025) were retrieved from the NCBI database, and the sequences shorter than 400 nt or containing degenerative positions were manually eliminated, resulting in a final dataset of 1052 sequences, including 13 Serbian sequences from this study, all from different host plants and different geographic regions worldwide. Genetic diversity was analyzed using DnaSP version 6.0 [52], which provided the number of haplotypes (h), haplotype diversity (Hd), number of variable sites (S) and nucleotide diversity (π) for the ITS region. Further analyses for haplotype composition and frequency were performed using PopART 1.7 software [53]. Nucleotide identities between sequences were calculated with MEGA 7.0 software [47], which allowed a comparison of sequence similarity and divergence. A haplotype network of all selected sequences and the Serbian cabbage and sunflower isolates was generated using the median joining network algorithm implemented in PopART [54].

### 4.6. Pathogenicity and Aggressiveness Testing

To prepare the inoculum for pathogenicity and aggressiveness studies, all studied isolates were subcultured on PDA and incubated for 4 days at 24 °C in the dark. Mycelial plugs with a diameter of 10 mm, cut 10 cm from the edge of the colony, served as the inoculum source in both experiments.

Pathogenicity was tested under laboratory conditions through the inoculation of cabbage and sunflower plants at the growth stage of 4 true leaves. The mycelial plugs were placed on wounded stem tissue, covered with a moist cotton swab and sealed with aluminum foil. For each isolate, 5 plants were inoculated. The control plants were inoculated using the same method, but with sterile pieces of PDA. The inoculated plants were incubated in a greenhouse with a photoperiod of 12 h (light and dark) and a temperature of 24 °C.

To determine the aggressiveness of 13 *S. sclerotiorum* isolates (12 from cabbage and 1 from sunflower), 6 plant species from the Brassicaceae family and sunflower were used in a greenhouse experiment (Table 3). The seedlings of all experimental plants were grown under greenhouse conditions and inoculated at the growth stage of 6–8 leaves. The second internodes of the plants were superficially wounded using a laboratory needle, and the mycelial plugs of the respective isolates were placed on the injured tissue. The inoculation site was immediately covered with a moist cotton swab and sealed with aluminum foil (Figure 1M). The plants of the control group were inoculated with sterile PDA fragments. The plants were incubated in a greenhouse under natural light with a 12 h photoperiod and temperatures of 20 ± 2 °C. The experiment was a completely randomized block consisting of 6 plants/plant species/isolate and was replicated three times. Disease severity was assessed three days after inoculation. Symptoms were rated using the scale established for this experiment: 0—no reaction; 1—necrosis length up to 0.5 cm; 2—necrosis length 0.5–1 cm; 3—necrosis length 1–2 cm; 4—necrosis length > 2 cm. An average disease severity index was calculated. For each isolate, a cumulative disease severity index was calculated as the sum of the average values for all hosts combined, and for each host, the cumulative disease severity index was calculated as the sum of the average values for all isolates combined.

### 4.7. Statistical Analyses

The growth rate data (colony diameter of the studied isolates) were verified for normality using the CKolmogorov–-Smirnov and Liliefors tests and processed by one-way ANOVA using Graph Pad Software 5.0 (USA). Mean values were compared using Tukey’s test at the *p* < 0.05 level of significance. Results are presented as mean daily growth rate in mm/day ± standard deviation (SD).

The recorded number of sclerotia per plate that failed the normality test was subjected to Kruskal–Wallis non-parametric statistical analysis, followed by Dunn’s multiple comparison test at the significance level *p* < 0.05, using Graph Pad Software 5.0 (San Diego, CA, USA). Data were expressed as means ± SD.

The ordinal data of the greenhouse inoculation experiment were pooled together and subjected to the Kruskal–Wallis non-parametric statistical test separately for each host plant. The medians of the disease severity index were compared using Dunn’s multiple comparison test at the significance level *p* < 0.05. Data were expressed as means ± SD.

## Figures and Tables

**Figure 1 plants-14-02478-f001:**
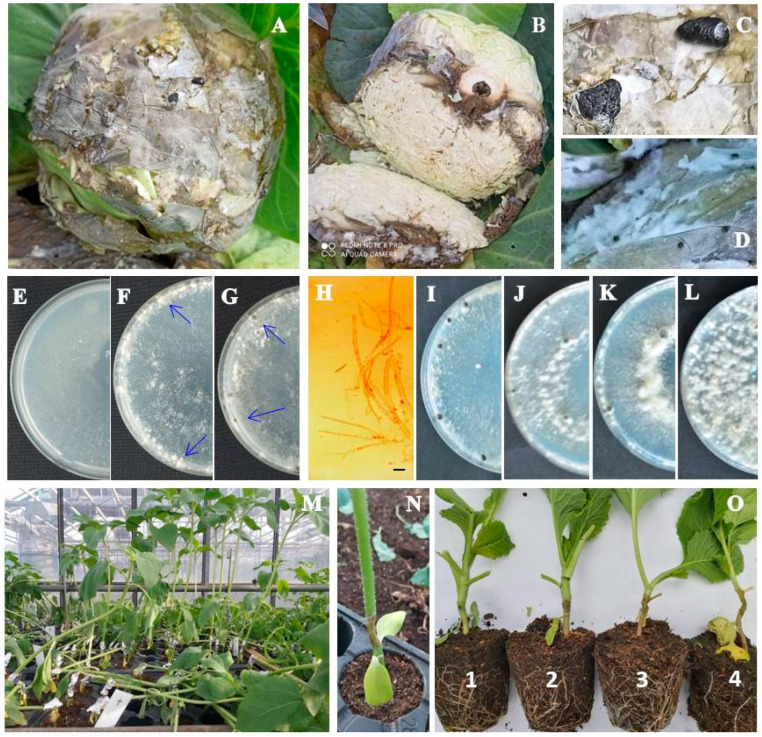
*Sclerotinia sclerotiorum*: (**A**,**B**) cabbage head rot (natural infection); (**C**,**D**) sclerotia and mycelium on infected tissue; (**E**,**H**) sparse, homogeneous mycelium of two-day old colony on PDA (bar = 20 μm); (**F**) white sclerotial initials in four-day colony on PDA (blue arrows); (**G**) fully developed sclerotia in six-day-old colony on PDA (blue arrows). (**I**–**L**) The 15-day-old colony morphology: (**I**) sparse floccose, (**J**) sparse–low dense floccose, (**K**) sparse–very dense wooly–floccose, (**L**) very dense, wooly. (**M**,**N**) White stem rot of artificially inoculated sunflower plants after three-day incubation. (**O**) Rating scale for disease incidence evaluation on cabbage plants: 1—necrosis length up to 0.5 cm; 2—necrosis length 0.5–1 cm; 3—necrosis length 1–2 cm; 4—necrosis length > 2 cm.

**Figure 2 plants-14-02478-f002:**
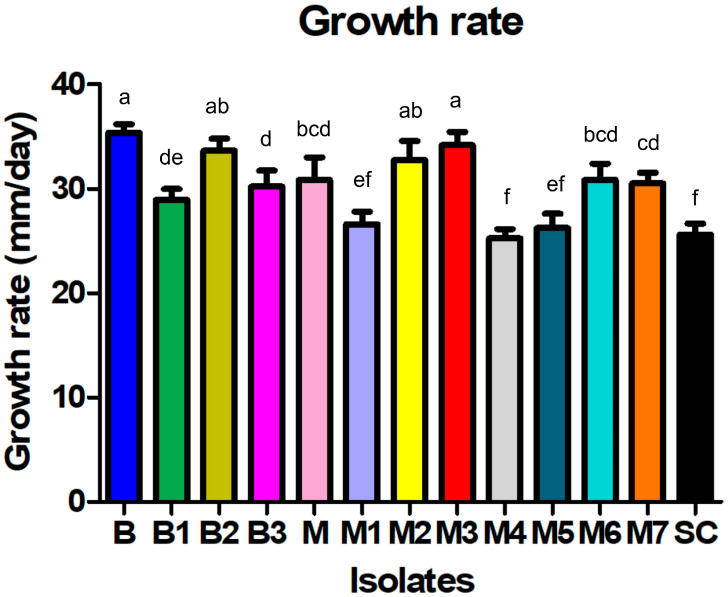
Growth rate of *Sclerotinia sclerotiorum* isolates on PDA at 24 °C in the dark. B—Isolates from the Begeč location; M—isolates from the Futog location; SC—isolate from sunflower. Bars represent mean values of 5 replicates. Error bars indicate standard deviation. Values labeled with the same letter do not differ significantly.

**Figure 3 plants-14-02478-f003:**
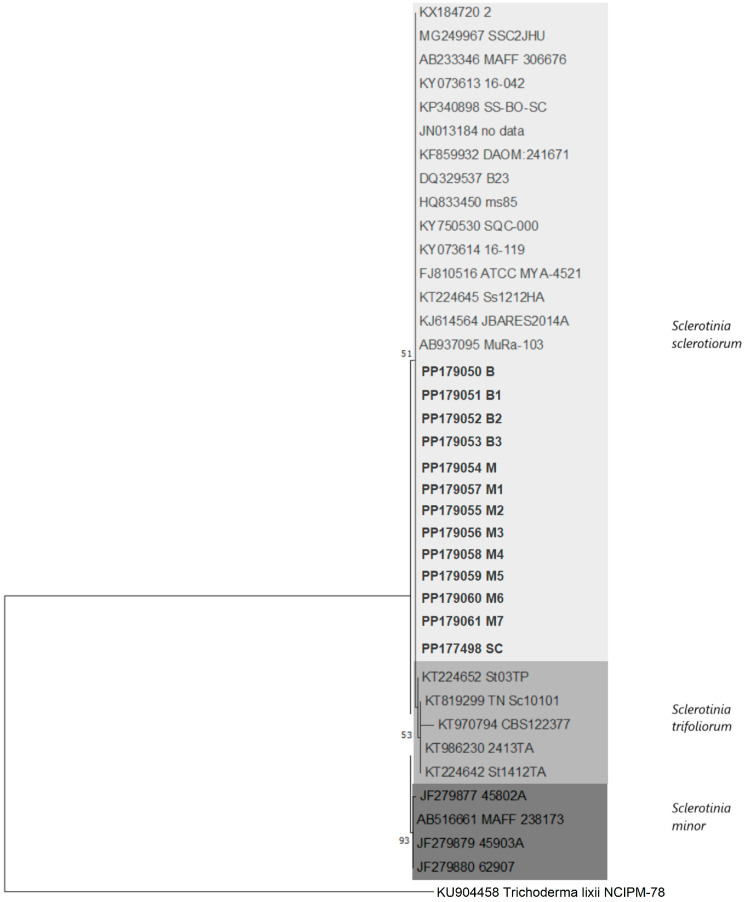
Maximum likelihood phylogenetic tree of the internal transcribed spacer rDNA sequences of 13 Serbian and 24 reference isolates of *Sclerotinia* spp., and the outgroup taxa *Trichoderma lixii*. The tree was generated in Mega 7.0 using Kimura’s two-parameter model. Bootstrap analyses were performed with 1000 replicates, and bootstrap values (>50%) are shown next to the corresponding branches. Serbian isolates are in bold.

**Figure 4 plants-14-02478-f004:**
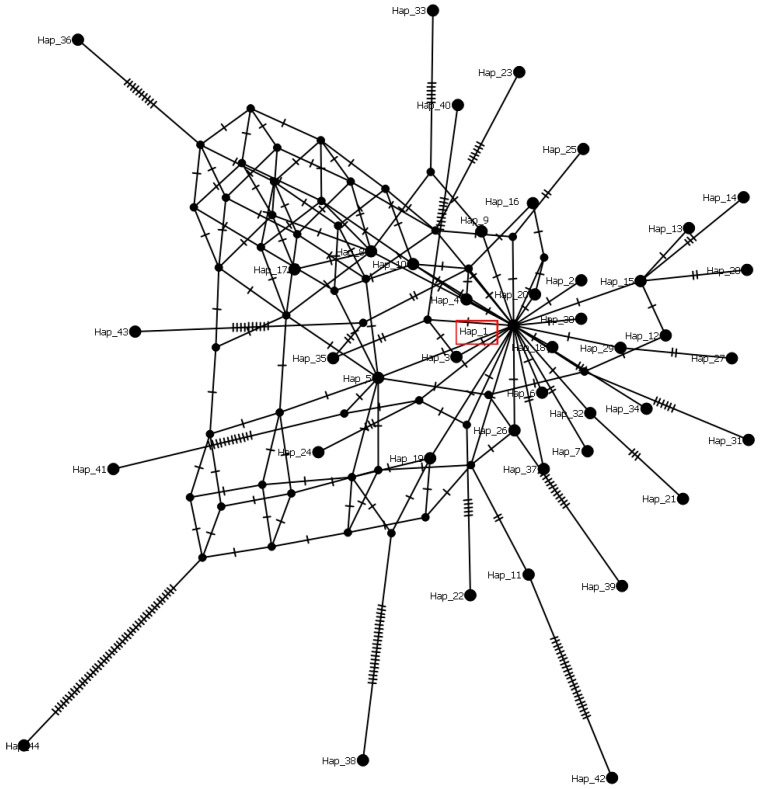
Median-joining network showing the phylogenetic relationships between haplotypes of *Sclerotinia sclerotiorum*. Black nodes represent median vectors required to connect existing haplotypes. Haplotype to which Serbian isolates belong is marked with a rectangle. The number of hatch marks on branches indicates mutational steps between haplotypes.

**Figure 5 plants-14-02478-f005:**
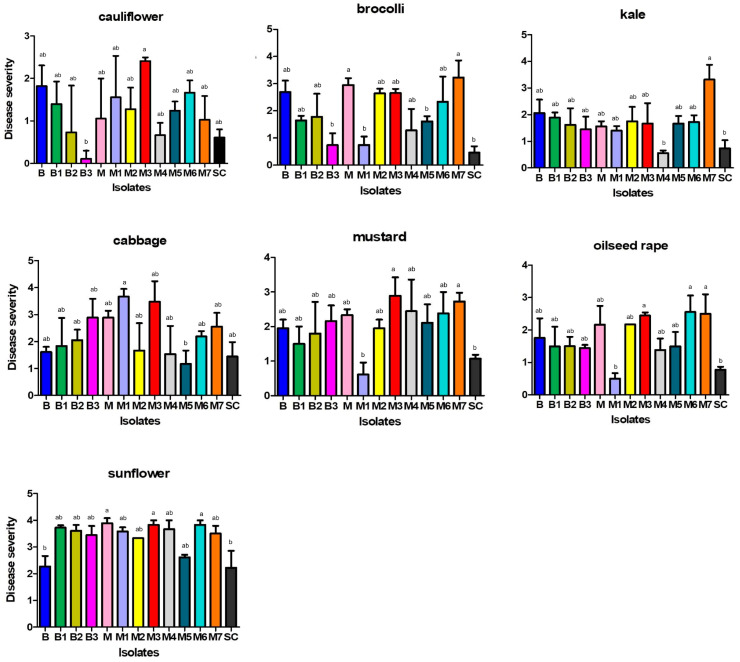
Disease severity on seven host plants caused by *Sclerotinia sclerotiorum*, rated using the following scale: 0—no reaction; 1—necrosis length up to 0.5 cm; 2—necrosis length 0.5–1 cm; 3—necrosis length 1–2 cm; 4—necrosis length > 2 cm. Error bars represent standard deviation. Values marked with the same letter do not differ significantly.

**Figure 6 plants-14-02478-f006:**
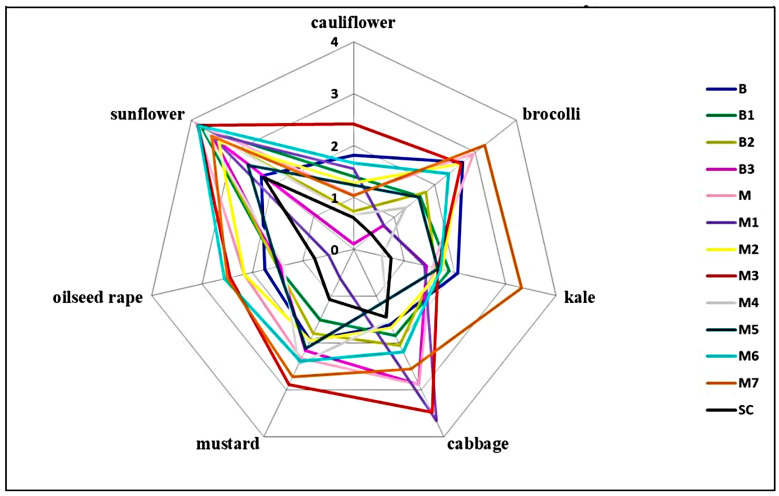
Comparative analysis of susceptibility levels in various host plants to *Sclerotinia sclerotiorum* isolates (B, B1-3, M, M1-7, SC) based on disease severity index calculated as an average symptom appearance rated using the following scale: 0—no reaction; 1—necrosis length up to 0.5 cm; 2—necrosis length 0.5–1 cm; 3—necrosis length 1–2 cm; and 4—necrosis length > 2 cm.

**Figure 7 plants-14-02478-f007:**
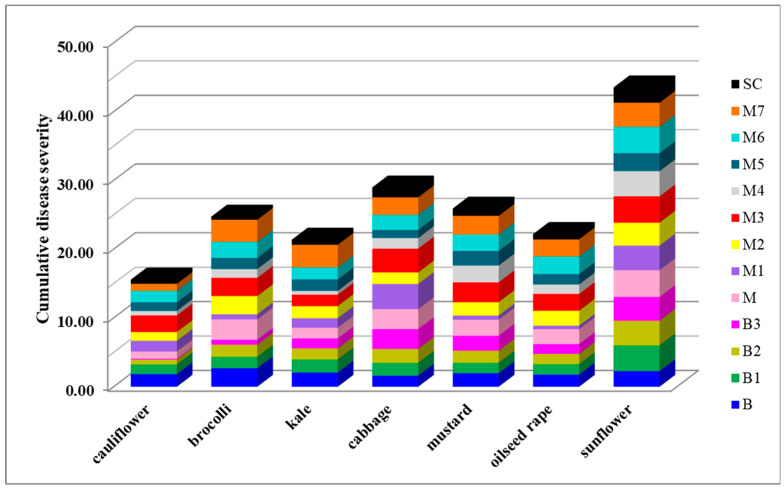
Sensitivity profiles of the host plants to *Sclerotinia sclerotiorum* isolates expressed as cumulative disease severity, calculated as the sum of the average disease severity index values (calculated as an average symptom appearance rated using the following scale: 0—no reaction; 1—necrosis length up to 0.5 cm; 2—necrosis length 0.5–1 cm; 3—necrosis length 1–2 cm; and 4—necrosis length > 2 cm) for all isolates combined.

**Figure 8 plants-14-02478-f008:**
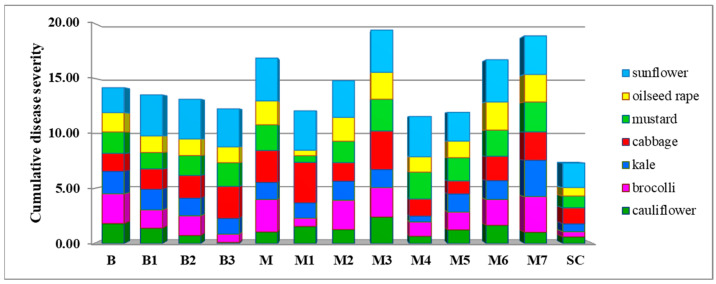
Aggressiveness profiles of the isolates of *Sclerotinia sclerotiorum* on different host plants expressed as a cumulative disease severity calculated as the sum of the average disease severity index values (calculated as an average symptoms appearance rated using the following scale: 0—no reaction; 1—necrosis length up to 0.5 cm; 2—necrosis length 0.5–1 cm; 3—necrosis length 1–2 cm; and 4—necrosis length > 2 cm) for all hosts combined.

**Table 1 plants-14-02478-t001:** Variability in colony appearance and sclerotia formation on PDA in *Sclerotinia sclerotiorum* isolates.

Isolate	Colony Appearance		Sclerotia			
	Color	Relative Density and Appearance of Areal Mycelium	Timing of Formation (day)	Average No. ± SD/Plate	Average Size (mm)	Arrangement
B	White	Sparse–very dense, Wooly–floccose	8	15.7 ± 1.5 cd *	5 × 3 (2–7 × 2–4)	Edge ring
B1	White	Sparse–low dense. Wooly	8	10.7 ± 0.6 de	4 × 3 (2–5 × 2–4)	Edge ring
B2	White	Sparse–low dense, Floccose	8	10.3 ± 0.6 de	6 × 3 (1–10 × 1–5)	Edge + middle rings
B3	White	Sparse–very dense, Wooly–floccose	6	15.7 ± 2.5 c	3 × 2 (1–5 × 1–3)	Edge ring
M	White	Dense, Wooly	8	10.7 ± 1.5 cde	5 × 3 (1–8 × 1–5)	Edge + middle rings
M1	White	Sparse, Floccose	8	10.7 ± 0.6 cde	4 × 3 (1–7 × 1–4)	Edge + middle rings
M2	White	Sparse, Floccose	9	9.7 ± 0.6 e	10 × 4 (2–7 × 2–6)	Middle ring
M3	White	Moderately dense, Floccose	6	12.7 ± 0.6 cde	5 × 4 (2–7 × 2–5)	Edge ring
M4	White	Sparse–dense, Floccose	8	16.7 ± 0.6 bc	6 × 3 (1–11 × 1–4)	Edge + middle rings
M5	White	Sparse, Floccose	6	23.7 ± 7.0 b	3 × 3 (1–5 × 1–4)	Edge ring
M6	White	Sparse–very dense, Wooly–floccose	8	16.7 ± 1.5 bc	5 × 4 (3–6 × 3–4)	Edge + middle rings
M7	White	Sparse–dense, Wooly–floccose	7	33.7 ± 1.5 a	6 × 3 (2–9 × 2–3)	Edge + middle rings
SC	White	Very dense, Wooly	13	2.7 ± 0.6 f	2 × 2 (1–2 × 1–2)	Edge + middle rings

* Values marked with the same letter do not differ significantly.

**Table 2 plants-14-02478-t002:** Isolates of *Sclerotinia sclerotiorum* recovered in this study and species from GenBank included in phylogenic analyses.

Species	Isolate	Acc No.	Host	Country	Literature
*Sclerotinia sclerotiorum*	2	KX184720	Cabbage (*Brassica oleracea* var. *capitata*)	Sri Lanka	[33]
	SSC2JHU	MG249967	Cotton (*Gossypium hirsutum*)	USA	[33]
	MAFF 306676	AB233346	Blueberry (*Vaccinium corymbosum*)	Japan	[33]
	16-042	KY073613	Shepherd’s purse (*Capsella bursa-pastoris*)	Korea	[33]
	SS-BO-SC	KP340898	Cabbage (*Brassica oleracea* var. *capitata*)	New Mexico	[33]
	-No data	JN013184	Fan Columbine (*Aquilegia flabellata*)	Italy	[33]
	DAOM: 241671	KF859932	No data	Canada	[33]
	B23	DQ329537	No data	Alaska	[33]
	ms85	HQ833450	Mulberry (*Morus alba*)	China	[33]
	SQC-000	KY750530	Chinese celery (*Oenanthe javanica*)	China	[33]
	16-119	KY073614	Cucumber (*Cucumis sativus*)	Korea	[33]
	ATCC MYA-4521	FJ810516	No data	USA	[33]
	Ss1212HA	KT224645	Caucasian clover (*Trifolium ambiguum*)	Poland	[33]
	JBARES2014A	KJ614564	Chinese chives (*Allium tuberosum*)	Korea	[33]
	MuRa-103	AB937095	Napa cabbage (*Brassica napa*)	Japan	[33]
	B	PP179050	Cabbage (*Brassica oleracea* var. *capitata*)	Serbia	This study
	B1	PP179051	Cabbage (*Brassica oleracea* var. *capitata*)	Serbia	This study
	B2	PP179052	Cabbage (*Brassica oleracea* var. *capitata*)	Serbia	This study
	B3	PP179053	Cabbage (*Brassica oleracea* var. *capitata*)	Serbia	This study
	M	PP179054	Cabbage (*Brassica oleracea* var. *capitata*)	Serbia	This study
	M1	PP179057	Cabbage (*Brassica oleracea* var. *capitata*)	Serbia	This study
	M2	PP179055	Cabbage (*Brassica oleracea* var. *capitata*)	Serbia	This study
	M3	PP179056	Cabbage (*Brassica oleracea* var. *capitata*)	Serbia	This study
	M4	PP179058	Cabbage (*Brassica oleracea* var. *capitata*)	Serbia	This study
	M5	PP179059	Cabbage (*Brassica oleracea* var. *capitata*)	Serbia	This study
	M6	PP179060	Cabbage (*Brassica oleracea* var. *capitata*)	Serbia	This study
	M7	PP179061	Cabbage (*Brassica oleracea* var. *capitata*)	Serbia	This study
	SC	PP177498	Cabbage (*Brassica oleracea* var. *capitata*)	Serbia	This study
*Sclerotinia trifoliorum*	St03TP	KT224652	Caucasian clover (*Trifolium ambiguum*)	Poland	[33]
	CBS122377	KT970794	Caucasian clover (*Trifolium ambiguum*)	Poland	[33]
	TN Sc10101	KT819299	Fenugreek (*Trigonella foenum-graecum*)	Tunisia	[49]
	St2413TA	KT986230	Caucasian clover (*Trifolium ambiguum*)	Poland	[48]
	St1412TA	KT224642	Caucasian clover (*Trifolium ambiguum*)	Poland	[48]
*Sclerotinia minor*	MAFF 238173	AB516661	No data	Japan	[33]
	45903A	JF279879	No data	Australia	[33]
	45802A	JF279877	No data	Australia	[33]
	62907	JF279880	No data	Australia	[33]
*Trichoderma lixi*	NCIPM-78	KU904458	Rhizosphere soil	India	[50]

**Table 3 plants-14-02478-t003:** The experimental plant species used to study the aggressiveness of isolates of *Sclerotinia sclerotiorum* from cabbage in Serbia.

Common Name	Latin Name	Cultivar	Cultivar’s Origin
Cabbage	*Brassica oleracea* var. *capitata*	Futoški	Futog, Novi Sad, Serbia
Cauliflower	*Brassica oleracea* var. *botrytis*	Incline	Sakata Seed Southern Africa (Pty) Ltd., Kempton Park, South Africa
Broccoli	*Brassica oleracea* var. *silvestris*	Merathon	Sakata Seed Southern Africa (Pty) Ltd., Kempton Park, South Africa
Kale	* Brassica * *oleracea*	Estoril	Sakata Seed Southern Africa (Pty) Ltd., Kempton Park, South Africa
Mustard	* Sinapis alba *	NS Bela	Institute of field and vegetable crops, Novi Sad, Serbia
Oilseed rape	* Brassica napus *	NS Svetlana	Institute of field and vegetable crops, Novi Sad, Serbia
Sunflower	* Helianthus annuus *	Labud	Institute of field and vegetable crops, Novi Sad, Serbia

## Data Availability

Dataset available on request from the authors.

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
