# Peer review of "Morphological, Pathogenic and Molecular Characterization of Sclerotinia sclerotiorum, the Causal Agent of White Rot of Cabbage (Brassica oleracea var. capitata), in Serbia"

_plants, 2025, doi:10.3390/plants14162478_

Round 1

Reviewer 1 Report

Comments and Suggestions for Authors

A reference should be included for the statement of lines 72-75. Is this information a personal communication?

Are the results presented as SEM in figure 2?

Figure 2 legend should indicate what B, M, and S isolates mean. In table 2, S is shown as SC.

Can the authors change the term aggressiveness to infection rate or disease severity in section 2.4? How can aggressiveness be measured?

In spite of necrosis, is there another parameter to evaluate disease severity?

Can the authors describe in section 4.1 whether there was an abnormality in the weather during September 2021 that could have caused this phenomenon?

If Sclerotinia sclerotiorum had never been reported as a phytopathogen in Serbia, how could it have been introduced (lines 272-276)?

The discussion section should clearly outline the limitations of this work.

Is there any perspective for this work? This should be mentioned in the discussion section.

A conclusion should be clearly stated at the end of the discussion section.

The format of some references is not homogeneous (e.g., 7, 18, 20, 23, and others).

I recommend the authors only include references to peer-reviewed articles.

Author Response

Response to Reviewer 1 Comments

1. Summary

We are grateful for your time to review this manuscript and useful suggestions which have improved our manuscript. Please find our answers below and the corresponding revisions/corrections in track changes in the re-submitted file.

2. Point-by-point response to Comments and Suggestions for Authors

Comment 1: A reference should be included for the statement of lines 72-75. Is this information a personal communication?

Response 1: The statement is describing starting point of our study so we have no reference to be included.

Comment 2: Are the results presented as SEM in figure 2?

Response 2: The results in Fig 2 are presented as mean daily growth rate in mm/day ± standard deviation (SD) as described in the Material and Methods section and indicated in the Fig 2 capture.

Comment 3: Figure 2 legend should indicate what B, M, and S isolates mean. In table 2, S is shown as SC.

Response 3: Thank you for pointing this out. We clarified the origin of the isolates and B, M and SC isolate codes were explained in Fig. 2. We also corrected S to SC in the Figure 2. (Lines 97-100 and 171-175 of the revised manuscript)

Comment 4: Can the authors change the term aggressiveness to infection rate or disease severity in section 2.4? How can aggressiveness be measured?

Response 4: Aggressiveness or virulence is common term used to describe quantitative differences in the ability of a pathogen to cause disease and it can be measured as explained in M&M. We suggest retaining the term aggressiveness in the manuscript.

Comment 5: In spite of necrosis, is there another parameter to evaluate disease severity?

Response 5: Our primary aim was to evaluate the relative susceptibility of different plant species from Brassicaceae family to the pathogen and to compare the aggressiveness of the pathogen isolates to different host species. Based on this objective, we chose lesion length as a reliable and widely used parameter for disease severity in similar host–pathogen systems and as the first visible plant response. In addition, lesion length is commonly used in similar experiments and has been shown to correlate well with pathogen aggressiveness and host susceptibility in many pathosystems, especially in controlled inoculation experiments.

Comment 6: Can the authors describe in section 4.1 whether there was an abnormality in the weather during September 2021 that could have caused this phenomenon?

Response 6: In August and September 2021, weather conditions in the Vojvodina region were typical for that time of year. Thus, the observed disease outbreak did not appear to be directly influenced by the weather conditions.

Comment 7: If Sclerotinia sclerotiorum had never been reported as a phytopathogen in Serbia, how could it have been introduced (lines 272-276)?

Response 7. As explained in lines 280-285 of the revised manuscript, Sclerotinia sclerotiorum presence has been previously documented in the country—particularly in the province of Vojvodina—on various host plants, especially sunflower [14]. The first observation of symptoms on cabbage, the host plant investigated in this study, occurred in 1997 (Mitrović, personal communication) and was later published in 2016 [17]. However, no subsequent research has been conducted to further investigate the population structure, distribution, or host range of the pathogen in Serbia. Our study documents the first occurrence on cabbage that goes beyond the general observation of occurrence and includes a detailed identification and characterization of the S. sclerotiorum cabbage population. This data is of great importance to cabbage growers and our study attempts to provide an answer to the question of which Brassicaceae crops are best suited for crop rotation.

Comment 8: The discussion section should clearly outline the limitations of this work.

Response 8. This request was accepted. The following change has been made: “Although the results of this study provide important insights into aggressiveness of S. sclerotiorum isolates from cabbage in Serbia, these tests were carried out under controlled conditions, which do not fully reflect the complex interactions between the pathogen, host plant, and environmental factors that influence disease development in agroecosystems. Nevertheless, this study provides a valuable foundation for understanding the presence and biological behavior of S. sclerotiorum in cabbage production, as well as guidance for future research aimed at improving sustainable disease management strategies. “ (Lines 388-395 of the revised manuscript).

Comment 9: Is there any perspective for this work? This should be mentioned in the discussion section.

Response 9: This suggestion was accepted. The following paragraph was included in the discussion section: “Pronounced phenotypic variability of S. sclerotiorum isolates, particularly in the aggressiveness and host specificity, suggests that molecular markers alone may not fully capture the pathogen’s adaptive potential. To unravel the mechanisms underlying phenotypic variability, future research should focus on multi-locus genotyping or whole-genome sequencing, coupled with long-term field surveillance to detect the emergence of highly aggressive or host-adapted strains.” (Lines 371-376 of the revised manuscript)

Comment 10: A conclusion should be clearly stated at the end of the discussion section.

Response 10. This suggestion was accepted. The last paragraph of the discussion section was changed to clearly stated conclusion as follows: The present study clearly revealed that even genetically homogeneous populations of S. sclerotiorum can display considerable phenotypic plasticity, likely driven by environmental conditions and host-specific selective pressures. The elucidation of pathogenic variability within this clonal population highlights the need for incorporating both molecular and phenotypic characterization in the development of integrated disease management strategies in Brassicaceae cropping systems.” (Lines 399-404 of the revised manuscript)

Comment 11: The format of some references is not homogeneous (e.g., 7, 18, 20, 23, and others).

Response 11. Thank you for pointing this out. We re-checked the format of references and corrected incorrect once.

Comment 12: I recommend the authors only include references to peer-reviewed articles.

Response 12: We fully agree with the reviewer that peer-reviewed references should be included whenever possible. However, in some instances, essential information relevant to the topic of this study—particularly concerning prior research conducted in Serbia—is not available in peer-reviewed journals. While we have made every effort to minimize the use of non-peer-reviewed sources and to prioritize journal articles, the remaining references of this kind have been included only where necessary to ensure the completeness and accuracy of the introduction and discussion.

Reviewer 2 Report

Comments and Suggestions for Authors

In 2021, white rot symptoms on cabbage were observed in 12 commercial fields in northern part of Serbia, and 12 representative isolates of S. sclerotiorum, forming white colonies and black sclerotia, were selected for characterization and comparison with isolate from sunflower, as the most important host plant in Serbia. The isolates showed significant variation in growth rate and sclerotia production, while ITS sequence analysis revealed complete nucleotide identity and all isolates grouped within the major phylogenetic clade of S. sclerotiorum. Despite the low diversity of global population of S. sclerotiorum, 44 haplotypes were identified with one predominant haplotype encompassing all Serbian isolates. When six Brassicaceae species (cabbage, cauliflower, broccoli, kale, mustard, and oilseed rape) and sunflower were inoculated, sunflower was found to be the most and cauliflower the least susceptible, while isolates from cabbage were generally more aggressive than those derived from sunflower. It is a very good work.

However,

  1. All the latin name should be italic.
  2. Line 57 and 273, "Sclerotinia scleroiorum" should be "S. sclerotiorum".
  3. "Based on the morphology and growth characteristics of the recovered isolates and the reproduced symptoms on inoculated cabbage and sunflower plants, the causal agent of the investigated cabbage disease was identified as S. sclerotiorum." should be moved to 2.3.
  4. In the section 4.6, how was disease severity assessed? please provided the formulationand incited reference.

Author Response

Response to Reviewer 2 Comments

1. Summary

We thank you for your time in reviewing this manuscript and for your useful suggestions that have improved our manuscript. Below you will find our responses below and the corresponding revisions/corrections in track changes in the re-submitted file.

2. Point-by-point response to Comments and Suggestions for Authors

Comments 1: All the latin name should be italic.

Response 1: We have accepted the suggestion and italicized all the names of fungal genera, but the names of botanical or fungal families are not italicized, following the examples found in other articles of the Plants.

Comments 2: Line 57 and 273, "Sclerotinia scleroiorum" should be "S. sclerotiorum".

Response 2: Agree. We have, accordingly, changed Sclerotinia sclerotiorum in S. sclerotiorum”. (Lines 57 and 280 of the revised manuscript).

Comments 3: "Based on the morphology and growth characteristics of the recovered isolates and the reproduced symptoms on inoculated cabbage and sunflower plants, the causal agent of the investigated cabbage disease was identified as S. sclerotiorum." should be moved to 2.3.

Response 3: Thank you for pointing this out. We agree with this comment. Therefore, we have change this paragraph to: Therefore, based on the morphology and growth characteristics of the recovered isolates, reproduced symptoms on inoculated cabbage and sunflower plants, sequence and phylogenetic analysis, the causal agent of the investigated cabbage disease was identified as S. sclerotiorum.” and moved it to 2.3. Section. (Lines 200-203 of the revised manuscript).

Comments 4: In the section 4.6, how was disease severity assessed? please provided the formulationand incited reference.

Response 4: Thank you for pointing this out. We agree with this comment and clarified the disease severity assessment method. The following change has been made: “Disease severity was assessed three days after inoculation. Symptoms were rated using the following scale established for this experiment: 0 – no reaction; 1 – necrosis length up to 0.5 cm; 2 – necrosis length 0.5-1 cm; 3 – necrosis length 1-2 cm; 4 – necrosis length >2 cm, and an average disease severity index was calculated.” (Lines 506-509 of the revised manuscript).

Reviewer 3 Report

Comments and Suggestions for Authors

The manuscript, titled "Morphological, Pathogenic and Molecular Characterization of Sclerotinia sclerotiorum, the Causal Agent of White Rot of Cabbage (Brassica oleracea var. capitata) in Serbia" deals with the investigation of Serbian subpopulations of a globally significant, polyphagous pathogenic fungus. Its occurrence on cruciferous plants is also a significant plant pathology problem, especially in oilseed rape, but its occurrence on head cabbage and other related vegetables is not very common.

The introduction presents in sufficient detail the results of similar studies related to Sclerotinia sclerotiorum infections, especially in cruciferous vegetables.

The materials and methods section has been compiled in sufficient detail. It combines classical mycological morphological studies with molecular biological taxonomic methods. It describes in detail the applied test methods and data processing methods. All of this enables the reproducibility of the performed tests. It determines the virulence differences between the individual Serbian S. sclerotiorum isolates and the susceptibility of the most important brassicas by artificial inoculation tests.

The presentation of the results is of a good standard and well illustrated. Regarding the results of the research, the pathogen responsible for the symptoms was clearly identified and its haplotype was also determined. Based on this, it was concluded that the Vojvodina subpopulations of the pathogen can be considered quite clonal. At the same time, it was also revealed that the isolate of the pathogen from sunflower showed less aggressiveness than the others isolated from brassicas.
In the case of Figure 2, I recommend that the isolation location of the individual isolates (B-S) be given in the explanation of the figure or in a separeted table. I also recommend that the manuscript be re-read for typographical errors correction (e.g., cucamber written instead of cucumber in several places).

The discussion and conclusion chapters provide a good summary of the results achieved and compare them with research on other infected plant species and fungal isolates. In total, the manuscript cites 54 closely related literature sources.

After the above suggested additions and improvements have been made, I propose publishing the manuscript as a scientific article.

Author Response

Response to Reviewer 3 Comments

1. Summary

We are grateful for your time to review this manuscript and useful suggestions which have improved our manuscript. Please find our answers below and the corresponding revisions/corrections in track changes in the re-submitted file.

2. Point-by-point response to Comments and Suggestions for Authors

Comments 1: In the case of Figure 2, I recommend that the isolation location of the individual isolates (B-S) be given in the explanation of the figure or in a separeted table.

Response 1: We clarified the origin of the isolates and the following change has been made: “From the collected symptomatic cabbage samples, 12 Sclerotinia-like isolates were obtained, 4 isolates from Begeč (B, B1, B2, and B3) and 8 isolates from Futog (M, M1, M2, M3, M4, M5, M6, and M7), one representative per cabbage production field, all of which showed a similar colony appearance, typical for Sclerotinia spp.” (Lines 97-100 of the revised manuscript). In addition, B and M were explained in Fig. 2. (Lines 173-175 of the revised manuscript).

Comments 2: I also recommend that the manuscript be re-read for typographical errors correction (e.g., cucamber written instead of cucumber in several places).

Response 2: Thank you for pointing this out. We re-read for typographical errors and corrected them. (Lines 468 and 517 of the revised manuscript).

Round 2

Reviewer 1 Report

Comments and Suggestions for Authors

Lines 72-75 should state that this is personal communication.

The legend for Figure 2 should clarify whether the results represent the mean and standard deviation, along with the sample size (n) for each treatment.

In spite of necrosis, is there another parameter to evaluate disease severity? The author's response should be included in the manuscript.

Can the authors describe in section 4.1 whether there was an abnormality in the weather during September 2021 that could have caused this phenomenon? The author's response should be included in the manuscript.

If Sclerotinia sclerotiorum had never been reported as a phytopathogen in Serbia, how could it have been introduced? The author's response should be included in the manuscript.

Author Response

Response to Reviewer 1 Comments

1. Summary

Thank you for investing patience and good will in reviewing our manuscript and useful suggestions which have clarified several points. Please find our answers below and the corresponding revisions/corrections in track changes in the re-submitted file.

2. Point-by-point response to Comments and Suggestions for Authors

Comment 1: Lines 72-75 should state that this is personal communication

Response 1: This suggestion was accepted and Mitrović, personal communication was added in the sentence. (Lines 72-75 of the revised manuscript).

Comment 2: The legend for Figure 2 should clarify whether the results represent the mean and standard deviation, along with the sample size (n) for each treatment.

Response 2: This suggestion was accepted. The following change in the legend for Figure 2 has been made: “Bars represent mean values of 5 replicates. Error bars indicate standard deviation.” (Lines 173-175 of the revised manuscript).

Comment 3: In spite of necrosis, is there another parameter to evaluate disease severity? The author's response should be included in the manuscript.

Response 3: Thank you for pointing this out. The following change has been made: To ensure methodological consistency and comparability, lesion length was selected as the primary indicator of susceptibility to S. sclerotiorum, a parameter well-established in the literature, especially in controlled inoculation experiments, and applicable across plant species with distinct anatomical and morphological features [5,6, 14,21,22].” (Lines 350-354 of the revised manuscript).

Comment 6: Can the authors describe in section 4.1 whether there was an abnormality in the weather during September 2021 that could have caused this phenomenon? The author's response should be included in the manuscript.

Response 6: This suggestion was accepted. The following change has been made: “The observed disease outbreak did not appear to be directly influenced by the weather conditions, since they were typical for that region and the time of year (data not shown).”

(Lines 418-420 of the revised manuscript).

Comment 7: If Sclerotinia sclerotiorum had never been reported as a phytopathogen in Serbia, how could it have been introduced (lines 272-276)?

Response 7. As explained in lines 280-286 of the revised manuscript, Sclerotinia sclerotiorum presence has been previously documented in the country—particularly in the province of Vojvodina—on various host plants, especially sunflower [14]. The first observation of symptoms on cabbage, the host plant investigated in this study, occurred in 1997 (Mitrović, personal communication) and was later published in 2016 [17]. However, no subsequent research has been conducted to further investigate the population structure, distribution, or host range of the pathogen in Serbia. The following sentence was included in the manuscript: Our study is the first comprehensive characterization of the S. sclerotiorum population affecting cabbage in Serbia that offers valuable guidance for cabbage growers, particularly in selecting appropriate Brassicaceae crops for crop rotation.” (Lines 289-292 of the revised manuscript).